# From Queries to Criteria: Understanding How Astronomers Evaluate LLMs

**Alina Hyk**[*,1]**, Kiera McCormick**[*,2,3]**, Mian Zhong**[3]**, Ioana Ciucă**[4]**, Sanjib Sharma**[5]**,**

**John F. Wu**[3,5]**, J. E. G. Peek**[3,5]**, Kartheik G. Iyer**[6]**, Ziang Xiao**[3]**, Anjalie Field**[3]

[1]Oregon State University [2]Loyola University Maryland [3]Johns Hopkins University
[4]Stanford University [5]Space Telescope Science Institute [6] Columbia University

## Abstract

There is growing interest in leveraging LLMs to aid in astronomy and other scientific research, but benchmarks for LLM evaluation in general have not kept pace with the increasingly diverse ways that real people evaluate and use these models in scientific inquiry. In this study, we seek to improve evaluation procedures by building understanding of how users evaluate LLMs. We focus on a particular use case: an LLM-powered retrieval-augmented generation bot for engaging with astronomical literature, which we deployed in the real world. Our inductive coding of 368 queries to the bot over four weeks and our follow-up interviews with 11 astronomers reveal how experts evaluated this system, including the types of questions asked and the criteria for judging responses. We synthesize our findings into recommendations for building better benchmarks, which we then employ in constructing a sample benchmark for evaluating LLMs for astronomy. Overall, our work offers ways to improve LLM evaluation and, ultimately, real-world interaction, particularly for use in scientific research.

## 1 Introduction

There is growing interest in the astronomy community in using large language models (LLMs) to accelerate scientific research, as evidenced by tools like AstroLLaMa (Nguyen et al., 2023; Perkowski et al., 2024) and Pathfinder (Iyer et al., 2024), which aim to democratize astronomy by enabling global researchers and enthusiasts to search the vast body of astronomical literature, synthesize information, and refine their ideas using AI as a source of inspiration (Krenn et al., 2022). However, leveraging this potential remains difficult without principled evaluation frameworks that are grounded in real-world user interactions. Existing datasets to evaluate LLMs for science are dominated by closed-form questions, such as multiple-choice questions, which do not reflect the increasingly diverse ways people seek to use LLMs  (Sun et al., 2024; Singhal et al., 2023; Guo et al., 2023; Hendrycks et al., 2021). The challenge of evaluating LLMs for astronomy and other scientific fields mirrors current challenges in LLM evaluation more generally: while increasingly powerful models perform indistinguishably well on benchmarks, this high performance does not directly translate into usability and reliability, as benchmarks often fail to represent realistic user interactions (Bowman & Dahl, 2021; Raji et al., 2021; Liang et al., 2023; Liao & Xiao, 2023).

In this work, we explore how people use and evaluate an LLM-powered bot for astronomy literature exploration, ultimately translating our analysis into recommendations for building benchmarks that are reflective of the evaluation strategies and criteria that users employ. The system we deploy is a retrieval augmented generation (RAG) LLM: a user sends a query to the bot, the query is compared against a large body of astronomy literature to retrieve the most relevant documents, and the retrieved documents and the user query are passed to an LLM to provide a response with citations to the user. Astronomy serves

---

[*]Equal contribution

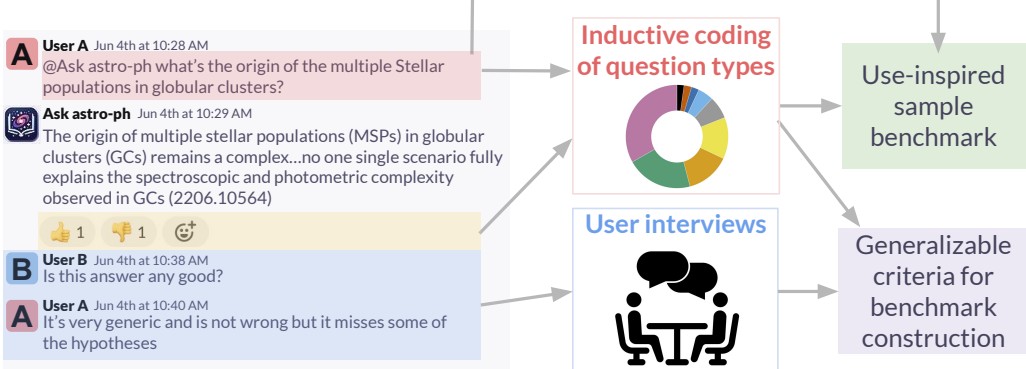

Figure 1: We deploy `@Ask-astro-ph` as a Slack bot and analyze resulting usage through inductive coding of user queries and follow-up interviews. We ultimately synthesize results into user-desired evaluation criteria, which are operationalizable in benchmark datasets.

as an ideal test domain for investigating users queries to LLMs for scientific research for several reasons. First, the field contains rich open-access literature that spans many decades of research. For example, NASA's Astrophysics Data System (ADS) hosts over 15 million resources encompassing virtually all literature used by astronomers (Accomazzi et al., 2015; 2022), creating a comprehensive knowledge base ideal for LLM applications. Second, short-term direct impacts of research on people are rare, reducing the potential harms that AI could cause. Finally, astronomy researchers constitute a well-defined user group, generally clustered around major research centers, which makes accessing potential users straightforward through collaborations and partnerships. Through our system deployment, we investigate two research questions motivated by our goal of understanding and improving LLM evaluation:

- **RQ1:** What types of questions do astronomers use to evaluate a bot for interacting with academic literature?
- **RQ2:** What criteria do astronomers judge in the bot's responses?

To investigate these questions, as summarized in Figure 1, we built and deployed the Slack bot `@Ask-astro-ph` and invited astronomers to interact with it in a group channel or via direct messages; over a span of four weeks, we logged user queries, emoji reactions, and user replies to the bot's responses. We then used an inductive coding approach to analyze the resulting 368 queries and conducted further follow-up interviews with 11 users. All users who interacted with the bot consented to make their de-identified interactions fully public as part of this study,[1] offering valuable data for further investigation.

Finally, we synthesize our analysis into specific recommendations on how this work can be leveraged to design human-centered evaluations of LLMs, and we release a sample gold benchmark for evaluating LLMs for astronomy[2]. Our sample benchmark consists of 40 question-answer pairs, where astronomers have written free-form answers with high-quality citations in response to real user queries. Thus, our sample data set is valuable for evaluating information retrieval as well as LLM response generation.

Although our work focuses on astronomy, we anticipate that our findings on LLM use and evaluation will generalize to other fields, particularly other observational sciences and semi-verifiable domains. Our overall approach of deriving evaluation criteria and ultimately constructing a benchmark grounded in real user testing is also easily repeatable, e.g., our astronomy-focused study serves as an initial low-risk proving ground, as opposed to disciplines like seismology or epidemiology, wherein deploying and evaluating LLMs

---

[1]https://huggingface.co/datasets/jhu-clsp/astro-llms-full-query-data
[2]https://huggingface.co/datasets/jhu-clsp/astro-llms-benchmark-dataset

is a riskier and more ethically fraught endeavor. We ultimately aim to improve alignment between user evaluations and benchmark datasets, with the long-term goal of broadly improving the usability of LLMs for science.

## 2 Related Work

Most progress in developing LLMs for scientific research has taken the form of benchmark datasets, which offer ways to evaluate model performance for related tasks, like medical question answering (Singhal et al., 2023) or mathematical problem solving (Hendrycks et al., 2021). Given the difficulty of creating benchmark datasets for scientific evaluation, which requires a high level of expertise, many datasets use automated construction methods, including extracting entities and relations from structured knowledge resources (Pampari et al., 2018; Pappas et al., 2018; 2020). However, these benchmark datasets often fail to reflect realistic use cases (Raji et al., 2021) and do not always correlate with human judgments (Tang et al., 2023; Fabbri et al., 2021), particularly for scientific inquiry which involves nuanced interpretation and domain-specific knowledge. For example, in task set-ups similar to ours, Gao et al. (2023) and Asai et al. (2024) evaluate LLMs along dimensions like fluency and citation correctness, but they do not justify if these metrics are important to users.

Some work has conducted more realistic evaluations of LLMs through user studies, most commonly for creative writing tasks, like scripts (Mirowski et al., 2023) or novels and fiction (Yang et al., 2022; Calderwood et al., 2020). A few studies have also considered research tasks. For example, Wang et al. (2024) investigate how researchers interact with LLMs, but they assign participants specific tasks rather than allowing open-ended exploration. In contrast, Zhao et al. (2024) take a more open-ended approach to study LLM use, but their initial work contains limited analysis and does not focus on scientific research. The absence of studies investigating how scientists naturally evaluate and use LLMs creates a significant gap in our understanding of effective evaluation frameworks for scientific applications.

While more reflective of real-world usage than curated benchmarks, user evaluations can be challenging to reproduce and do not facilitate efficient development, as they are too time-consuming and labor-intensive to conduct for every small change in a model. The non-determinism of LLMs can make reproducibility challenging even with static benchmarks (Blackwell et al., 2024), though improved output consistency is generally attainable through techniques like greedy decoding or lower temperature values (Renze, 2024; Patel et al., 2024; Song et al., 2024). Scientific domains like astronomy face additional challenges, including the need to integrate numerical data with textual knowledge and handle specialized terminology (Manakul et al., 2023), as well as mitigating hallucination risks which could compromise research reliability (Li et al., 2023; Zhang, 2023).

Overall, the continued utility of benchmarks for standardized evaluation, combined with their failure to reflect realistic use cases of LLMs for scientific research, motivates our work. We take an open-ended exploratory approach to identifying how users evaluate LLM-based systems in astronomy, and we translate our analysis into principled criteria to guide future evaluations in scientific research contexts.

## 3 Methodology

### 3.1 System Deployment and Data Collection

We constructed an LLM-powered RAG system titled `@Ask-astro-ph` as a bot on Slack. When a user queried the bot, it first retrieved the top 5 most relevant papers with the highest cosine similarity using `bge-small-en-v1.5` embeddings (Xiao et al., 2024). The retrieval was based on a dataset of 300,000 arXiv astrophysics papers posted on or before July 2023 (Perkowski et al., 2024). ArXiv serves as a strong data source as surveys have shown 93-94% of physicists and astronomers use arXiv with 100% usage in astrophysics (Narayana & Bhandi, 2022). The abstracts, conclusion sections, and metadata (arXiv IDs and years) of the retrieved papers were concatenated with a prompt (Appendix A) and the initial user query and passed to `gpt-4o` to produce an answer for the user.

We deployed `@Ask-astro-ph` on the internal Slack workspace for the Space Telescope Science Institute (STScI) and recruited study participants to interact with the bot. STScI conducts astronomy research and operates space telescopes for NASA Astrophysics, employing $> 150$ science staff who spend 0% to 50% of their time doing research. Participants electronically signed a consent document and fill out a demographic questionnaire (Appendix B). As exemplified in Figure 1, they were invited to a private Slack group channel, where their posts were sent as queries to `@Ask-astro-ph`. To provide feedback, users could click on pre-populated thumbs-up/down emojis and comment on the bot's replies to any user's queries. We collected data from May 29, 2024 to June 24, 2024, during which time 368 total queries were sent to the bot. 35 out of 43 participants who signed up sent at least one query.

## 3.2 Inductive Data Coding and Follow-up Interviews

We analyzed the question types of the queries via inductive coding. First, four authors from astronomy (2), NLP (1), and psychology (1) independently coded 20-50 queries. We emphasized labeling the type of knowledge or evaluation solicited by the user, rather than how a model might answer the question. As users also sometimes repeated the same question to the bot with different exact phrasing, annotators also labeled if a question was a re-ask of a previous question. The four annotators then reconciled and discussed the produced labels and co-coded 100 queries, with multiple iterations of revising the scheme to finalize a scheme and reach consensus. The remaining 267 queries were first coded independently, with $\geq 3$ annotators per query, where $\geq 2$ of the annotators were astronomers. Krippendorff's alpha for the independent coding was 0.51. The sources of disagreement were often driven by the diversity of backgrounds of the annotators: for example, astronomers better distinguished which types of queries should be considered unresolved topics in astronomy, whereas other annotators were more familiar with stress-testing strategies. The final label is based on a majority vote of annotators, and in cases of disagreement, we discussed options until agreement is reached. These contested queries were labeled as such with notes on the reasons for their final label to support replicability.

We conducted follow-up semi-structured interviews with 11 study participants out of 19 invitations to further understand their evaluation practice. We selected interviewees who (1) sent the most number of queries, (2) provided the most reply comments, (3) received the most thumbs up from the answers to their queries and (4) likewise but received thumbs down.[3] Despite some refusals, our final set had at least 2 people from the top 5 for each category. Each interview was conducted by 1 of 3 authors: they began with a fixed script, but were permitted to deviate as needed. During the interview, the interviewer encouraged the interviewee to talk through a few examples of their queries. All interviews were recorded and transcribed. Four authors independently coded 1-2 interviews and consolidated identified themes to determine an initial coding scheme. They co-coded 3 interviews to establish a mutual understanding of the scheme and conducted several iterations of independently coding the remaining interviews, updating the scheme, and re-coding interviews. Overall our study has a similar number of participants as related studies focused on user behavior and follows standard practices for data analysis (Lazar et al., 2017; Liao et al., 2020; Zhou et al., 2022; Mirowski et al., 2023; Zhou et al., 2022; Feng et al., 2025).

# 4 Results

## 4.1 RQ1: What types of questions do scientists use to evaluate a bot for interacting with academic literature?

Overall, our study reveals a variety of question types, which go beyond factual questions typically included in benchmark datasets. Our analysis also shows that although the bot was designed for a specific use case (i.e., interacting with academic literature) and a specific

---

[3]In practice, criteria (1) and (2) identified active users, while criteria (3) and (4) selected for users who only sent a few queries, i.e. the user with the highest percentage of negatively rated queries is someone who sent 1-2 queries that all received thumbs down, whereas users who sent many queries typically had more mixed ratings of responses.

user group (i.e., astronomy researchers), there is significant diversity in the types of queries and interaction patterns among users.

### 4.1.1 User Queries

From Table 1, the most common question type[4] was *Knowledge Seeking: specific factual* (e.g., "What is the most massive known spiral galaxy in the universe?"), followed by *Deep Knowledge* requiring higher-order reasoning, such as producing a summary, speculation or synthesis of results across papers (e.g., "What are the most promising subfields of astronomical research for new discoveries?").

Although we focus on user perspectives rather than the performance of this particular system, Table 1 reports user ratings to each query type for completeness and to account for ways users may adjust their evaluation strategies based on bot performance. In general, bot responses were rated more positively than negatively, except for queries that involved specific bibliometric components (🟠 🔴 16% of total queries). The bot was not designed for this use case as existing tools already support this search type (e.g., ADS, Google Scholar).

| | % of queries | thumbs down | thumbs up |
|---|---|---|---|
| 🔴 KS: specific factual | 33% | 28% | **31%** |
| 🟢 Deep Knowledge | 21% | 25% | **39%** |
| 🟠 BS: topic | 14% | **49%** | 8% |
| 🟡 KS: broad description | 13% | 6% | **52%** |
| ⚪ Stress Testing | 7% | 27% | 27% |
| 🔵 Bot Capabilities | 5% | 16% | **28%** |
| 🔵 KS: procedure | 2% | 11% | **56%** |
| 🔴 BS: specific paper or author | 2% | **56%** | 0% |
| ⚫ Unresolved Topic | 2% | 0% | **75%** |

Table 1: Percent of total queries inductively coded as each question type, and the percent of bot responses to queries of the specified type that had net thumbs up/down ratings. They do not sum to 100% as many responses had no ratings. "KS" = "Knowledge Seeking" and "BS" = "Bibliometric Search"

Figure 2 provides deeper insight by visualizing categorized queries for each user sequentially.[5] Almost all users used a range of question types, with distributions varying greatly by user. Some users first queried with *Knowledge Seeking: specific factual* type (e.g., 3, 19, 23), whereas others started with harder *Unresolved Topic* questions (e.g., 17, 32).

Similarly, question types were also well-distributed across users. For example, *Stress Testing* and *Unresolved Topic* questions did not occur as a burst of queries from a single user but rather were recurring strategies across multiple users. Moreover, users were often persistent in their querying strategy regardless of the bot's responses: although the bot consistently performed poorly on bibliometric questions, users persisted in asking them (e.g., 13, 17, 26). In the long term, we speculate that users would adapt to the system's strengths, but in a shorter-term deployment, we can surmise that users' queries were more driven by their initial evaluation strategies and use cases than the performance of this specific system. One area where users show initial adaptation is when users tried rephrasing a previous question (unfilled shapes in Figure 2), which occurred 22 times. Among re-asked questions where the bot's original response received a rating, in 73% of cases the rating was thumbs-down.

### 4.1.2 Follow-up Interviews

Follow-up interviews allow us to more deeply probe why users asked these questions. We identified 4 common strategies motivating these queries:

---

[4]We describe each coding category and an example from the data in Table 4 (Appendix C)

[5]In Appendix E, we show the same data temporally instead of sequentially.

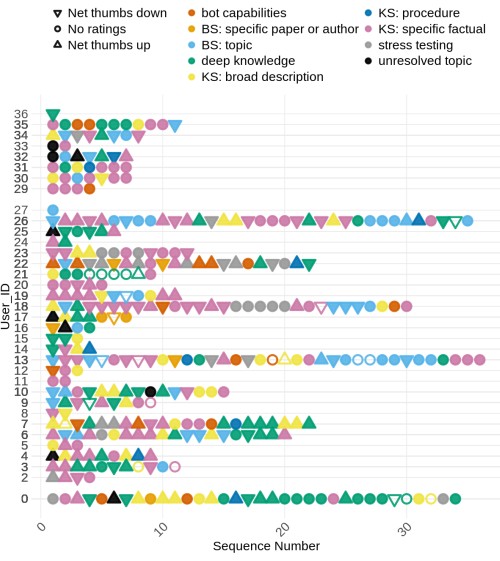

Figure 2: Question type and rating of bot response for each user.

**Query bot about things they know**   9 out of 11 users referenced querying the bot about things they know, primarily their specific area of expertise. An example is probing the bot for a question they knew was answerable from one of their recent publications: "So I knew[. . . ] I had a paper that was published during this period[. . . ] That it did have data available that was on this topic" {P9}. In other cases, users did not have a specific answer in mind, but were confident in their ability to judge a clearly incorrect answer.

**Trying out bot on actual use cases**   Surprisingly, even in this time-limited deployment, three users described queries as inspired by actual use cases, typically knowledge-seeking or bibliometric queries. One user mentioned querying an unfamiliar concept heard in a discussion group {P13}. Another described the motivation behind a query as: "This happened because I was explaining the same concept to a student.[. . . ] I sent some references back to them, and I was like, well, let's see if the bot can do better than I do." {P10}

**Nuanced, ambiguous, or difficult questions**   Other evaluation strategies included querying questions that were nuanced or difficult and may not have specific factual answers. These queries were typically coded as *Deep Knowledge* or *Unresolved Topic*, and {P4} described them as motivated by the scientific research process: "that's something really important to me, and something that I tried to emphasize when I work with students is that you know there's never a clear cut answer to anything."

**Trying to trick bot / probe for expected failures**   Finally, users attempted to trick the bot. In addition to off-topic queries like favorite food flavor, users also employed more nuanced approaches that targeted astronomy knowledge specifically, "And then I went a little on the ridiculous route, trying to see like well, if I post a question that is obviously wrong, would it still answer" {P7}, or targeted potential failures based on their knowledge of the bot's data sources: "There are the April Fool's articles that are always present there, and there was one about Wolf 359. So I wanted to see if it would pick that up." {P22}

### 4.2   RQ2: What criteria do scientists judge in the bot's responses?

In addressing our second research question, we focus primarily on results from follow-up interviews, as we found user thumbs up/down ratings insufficiently detailed to understand the desired evaluation criteria. Interviews revealed 5 commonly articulated dimensions:

**General correctness of response**   Unsurprisingly, the primary evaluation criterion that all users prioritized was overall correctness, though this criteria involved multiple sub-dimensions beyond general accuracy of terms and definitions. Users cared about the quality of citations, typically seeking for them to be relevant and recent, with mixed opinions on how the model should weigh citation counts. One user criticized more obscure references: "it would return some obscure reference that maybe had some of the words in the abstract, but was not necessarily a good reference to cite" {P26}, while another found the bot identifying previously unseen citations useful: "It would give me references that I hadn't read before[. . . ]so I would open those" {P4}. {P4} also emphasized the importance of the bot accurately reflecting the retrieved references: "I think the the most important thing to

me for a scientific response with a citation is that[. . . ] the content of the response that comes from the citation is accurately relayed[. . . ] if you can find a relevant citation and then you summarize it wrong, that's in some ways more insidious."

**Hedging and caveating of answers**   Eight users judged the model's abilities to qualify responses focusing on two areas: expressions of confidence and providing reasons for denying user requests. Users noted that the responses were often expressed confidently even when they were wrong: "One concern is when it does go wrong. It's hard to discern that it was wrong. It's very good at coloring things with a lot of confidence" {P34}. While prior investigations have similarly found that LLMs are prone to overconfidence and unreliable epistemic markers (Zhou et al., 2023; 2024), two of our users {P34, P4} identified this characteristic as a particular concern in the context of scientific research: "overconfidence is a bad thing to have as a researcher" {P4}. Regarding the second area, the deployed system was fairly conservative and frequently refused to answer queries, which users found frustrating, particularly because the bot failed to explain refusals: "I cannot answer, was responded a lot, and I kind of wish it had a little bit more of a qualifier to it...Can you not answer because it's not an astronomy question? Can you not answer because it's not in your source information?" {P22}. This frustration with unexplained denied requests reflects previous findings (Wester et al., 2024).

**Response specificity and clarity**   In addition to whether the bot response contained appropriate qualifiers, users often commented on the specificity of the responses, and noted that an answer could be factually correct but not informative: "I'm pretty sure this answer is accurate, but doesn't really tell you anything" {P22}. Users specifically used the word "generic" to describe these responses: "But that's like very generic, very generic answers" {P10}. Users otherwise generally did not comment on the response style, with a few suggesting that the language fluency was good enough not to be noteworthy.

**Correct interpretation of user query**   Another criterion that users evaluated in the model was its ability to correctly interpret the user query, which included inferring from context and understanding user intent. One user offered an example: "But somebody asked what's the typical radius of a $10^{15}$ solar mass cluster. And an astronomer would immediately know $10^{15}$ solar mass clusters: this person is talking about galaxy clusters, not about clusters of stars or other kinds–there are lots of uses of the word 'cluster' in astronomy–and the bot did not even reply until you were careful about it" {P9}. Another described the idea more generally in expressing what they liked about a different AI model (Claude): "It was as though it really understood the question better than[. . . ]the other programs did in terms of understanding what is exactly that this person wants from this question" {P36}. Users also described rephrasing queries or attempting to adapt to the query types that the bot responded best to. While from the user perspective, this involved a learning curve in understanding how to phrase queries, from a developer perspective, reducing the need for users to carefully phrase queries would be preferred.

**Broader Impact**   Finally, users also considered how such LLM tools might impact on their workflow, astronomy research practices, or society as a whole. Most commonly users evaluated whether the bot saved them time or work, e.g., {P9} sought "a quicker start" into literature exploration, and {P12} did not interact with the bot much because, "I realized I wasn't saving myself any work." Other users praised or looked for elements of discovery or surprise: "I had a really good time playing with that and discovering papers that I had never seen" {P4}. One user further considered the impact on the broader astronomy research: "[Does] using these agents make us better researchers make us more honest researchers?" and society as a whole: "there's a creeping feeling of like, what am I doing like? Who is this for? Who is this really benefiting?" {P12}. Recent efforts to benchmark LLMs have included criteria intended to capture possible societal impacts, such fairness or toxicity (Liang et al., 2023), though this criteria cannot fully answer questions like "Who is this really benefiting?".

# 5 Building Better Benchmarks: Guidelines and Construction

In this section, we synthesize results from §4.1 and §4.2, showing how inclusion of diverse query types and evaluation metrics are necessary for enabling benchmarks to capture users' desiderata. We further demonstrate the feasibility of implementing these guidelines by constructing a sample benchmark for evaluating LLMs for astronomy driven by our data.

In Table 2, we map the query types identified in §4.1 to the desired evaluation criteria identified in §4.2, offering concrete guidance on the types of queries to include in benchmarks in order to capture desired evaluation criteria. Our work suggests that existing benchmarks using closed-form questions (examples: Sun et al. (2024); Guo et al. (2023); Hendrycks et al. (2021)) are insufficient to support the broad range of criteria that users care about in using LLMs for science. This type of question is fundamentally limited to a subset of *Knowledge Seeking: specific factual* and only capable of capturing a narrow range of *General Correctness of Response*.

While the criteria in §4.2 and Table 2 more closely capture what scientists evaluate in models, one difficulty is how to implement them effectively. Asking annotators to rate model responses along these dimensions is straightforward, but relying on human annotation for every evaluation cycle does not facilitate efficient development. In the "Metric" column of Table 2 we suggest more automated approaches. For example, epistemic markers like "maybe", "might", or "definitely" are indicators of hedging and caveating (Zhou et al., 2023; 2024), and if retrieved evidence actually supports the model response is possible to evaluate through natural language inference (entailment) models (Gao et al., 2023), as has been implemented in systems for automated fact-checking (Thorne et al., 2018; Samarinas et al., 2021). We could partially capture all criteria by comparing model outputs to ground-truth paragraphs, using metrics such as n-gram overlaps (e.g., ROUGE) and BERTScore (Zhang* et al., 2020), though a single score would not be sufficient for providing nuanced feedback to developers. Asai et al. (2024) use LLM judgments for a range of criteria that has some overlap with our proposed dimensions, suggested that LLM judgments may be able to provide more nuanced evaluations.

| Evaluation Criteria | Query Type | Metric |
|---|---|---|
| **C1: Correct Interpretation of User Query** | 🟣🟢⚪🟡🔵⚫ | - Consistency under query paraphrases
- Closeness to human-written response |
| **C2: General Correctness of Response**
Correctness of Terms/Definitions
Quality of Retrieved Citations
Good Match of Response and Citation | 
🟣🟡
🟣🟡
🟣🟡🟢 | - Closeness to human-written response
- Exact match of closed-form answers
- Comparison with human-identified citations
- Natural language inference/entailment |
| **C3: Hedging and Caveat of Answers**
Level of Model Uncertainty
Explanation of Requests Denial | 
🟢🟡⚫
⚪🔵 | - Epistemic markers (e.g., "maybe", "definitely")
- Closeness to human-written response |
| **C4: Response Specificity and Clarity** | 🟢🟡🟣🟡🔵 | - Closeness to human-written response |
| **C5: Broader Impact** | N/A | - User studies |
| Note | 🟣 KS: specific factual 🟢 Deep Knowledge 🟠 BS: topic
🟡 KS: broad description ⚪ Stress Testing 🔵 Bot Capabilities
🔵 KS: procedure 🔴 BS: specific paper or author ⚫ Unresolved Topic |

Table 2: Recommended query types for capturing each user-desired evaluation criteria, as well as proposed metrics for facilitating fully automated evaluation.

While Table 2 offers concrete ways to capture evaluation criteria through inclusion of suitable query types and automated metrics, we do not identify ways to measure *Broader Impact* through benchmark datasets. This criteria is better captured through user studies and pilot deployments than static datasets.

## 5.1 A Sample Benchmark for Astronomy LLM Evaluation

We enact these recommendations in constructing a sample benchmark dataset consisting of 40 real user-LLM queries drawn from our primary dataset and gold answers with references

written by astronomers.[6] To our knowledge, this benchmark is the first evaluation dataset for astronomy that combines real user queries with expert-curated answers from astronomers.

To develop the benchmark, 7 astronomers read through the user queries and selected questions suited to their expertise. They then wrote an answer to each question and searched the literature to provide supporting references, typically 2 to 5 arXiv papers. Each question required about 30 minutes to answer, making the process time-consuming. One astronomer then read through all of the queries and answers to check for correctness. The chosen queries trace the frequency distribution of our primary dataset: approximately 55% are *Knowledge Seeking: specific factual*, 22% are *Deep Knowledge*, and 17% are *Knowledge Seeking: broad description*. Moreover, our benchmark retains the original format of the questions instead of standardizing or reformulating them, as this better reflects the patterns of interaction in the real world. Although this careful curation limits our initial dataset size, it establishes a pathway for creating high-quality question-and-answer pairs to evaluate LLMs in astronomy. The inclusion of free-form text responses and references allows for evaluating both generation and retrieval components for the proposed criteria on the quality and clarity of the response as well as the quality of retrieved citations.

We validate the quality of this benchmark and its usability for automated evaluation by comparing scores over the benchmark with users' thumbs up and down ratings. More specifically, for each query, we pass the human-written response and the original response generated by @Ask-astro-ph to gpt-4o-mini and prompt the model to generate a score between 0 and 1 reflecting the closeness of the two responses, with 1 meaning that the responses exactly match (Appendix F). Pearson's correlation between these LLM-generated scores and the thumbs-up/down counts from the primary dataset is 0.8239. This strong correlation suggests that automated evaluation using our sample benchmark is highly indicative of user satisfaction.

## 6   Discussion and Future Work

While we focus primarily on synthesizing individual user evaluations into automated benchmarks, our study further reveals alternative avenues for user-centric evaluations. As discussed in §3, we deployed the bot in a Slack space where users could interact with it in a group channel or through direct message. While some of the goals and outcomes of creating a shared channel were to encourage engagement, the channel also unexpectedly revealed ways users co-evaluated the bot. Users commented on each other's queries, offering direct advice on how to phrase queries or opinions about the bot's accuracy. Emoji reactions also reveal co-evaluation: out of 345 total recorded reactions, in 101 (29%) instances, the reaction was made by a different person than the initial query (typically the original querier had reacted as well). Similarly, in follow-up interviews, all participants except one referenced reading other users' queries without the interviewer prompting them. These interactions suggest collaborative practices could be potentially powerful tools for evaluating LLM-powered bots, which support several research initiatives that have considered the role of collaboration in data labeling (Muller et al., 2021; Kuo et al., 2024), and the social nature of Slack (Avula et al., 2018; 2019; 2022; Wang et al., 2022).

Our study further reveals high variance in how users interacted with the bot (Figure 2). While we do not find notable differences in how people of different genders interacted with the both, we do see differences by year of PhD completion, which is correlated with age: more recent graduates tended to ask more stress-testing and bot capabilities questions. Differences by year of degree completion / age are also evident in how users rated the bot: the bot's responses to queries from younger / more recent graduates were rated significantly more positively (Appendix D). These results emphasize that diverse groups people are needed to design evaluation schemes to ensure widespread usability, even within a relatively small community like astronomy researchers. Benchmark datasets are typically constructed

---

[6]We provide the entire 40 questions-and-answers benchmark as supplementary material. In future work, we plan to expand the dataset by obtaining expert-curated answers to a more significant subset of the remaining 325 questions, aiming for 100 for our second release, and we also intend to write paraphrases of queries and answers to support more variance in style.

by annotators who label data independently; they could be constructed collaboratively, for example, through participatory design workshops, which have been used to integrate broader perspectives into AI development in other ways (Brown et al., 2019; Katell et al., 2020).

Overall, the rapid development and adoption of LLMs, as well as the documented interest in leveraging them to advance scientific research, underscores an urgent need for improved evaluation schemes that are holistic and user-centric in this domain. Our work offers specific recommendations on how to improve current evaluation practices, and we expect many of our results to generalize from astronomy to other domains. We ultimately aim to broaden the usability of LLM-powered tools and in-turn, participation in scientific research.

## Ethics Statement

**IRB Review**    All protocols, including consent and procedure for de-identified data sharing, were approved by the Institutional Review Board (IRB) at Johns Hopkins University.

**Research Positionality**    We acknowledge that our academic backgrounds have influenced our perspectives on this topic. The authors of this work include 5 astronomers, 3 of whom are employed by the institute where the study took place. The emphasis on astronomy in this study reflects the interests and expertise of these authors, particularly in understanding the potential applications of LLMs in astronomy research. The remaining authors have backgrounds in NLP, HCI, and psychology.

**Limitations**    The primary limitation of our work is its focus on a small-scale user group in a specific domain. The astronomers we recruited were from a single institute, are currently based in the U.S. (although not all astronomers are originally from the U.S.), and already have PhDs, as opposed to students or amateur astronomers. Further, users opted-in to participate in the study, suggesting our user group may not capture the viewpoints of people without any interest (positive or negative) in AI tools. As STScI employs a broad range of astronomers with diverse backgrounds and areas of scientific expertise, we do expect our population to be representative within these criteria. Nevertheless, in order to achieve the goals of broadening the usability of AI tools, future work needs to proactively engage with a broader set of users, including incentivizing participation from less interested ones.

Our study is also limited in that it focused on evaluating one LLM-powered bot for a specific use case. People may use expanded evaluation strategies with a different tool, and their desired criteria may shift over longer-term use of a system integrated in their workflow. While we anticipate that many of the findings generalize beyond the specific astronomers who interacted with the tool and beyond astronomical research to other disciplines, especially those in semi-verifiable domains and/or observational sciences, we cannot conclusively assume generalizability.

## Acknowledgments

This work was conducted in part through a JSALT summer workshop hosted by the Center for Language and Speech Processing at Johns Hopkins University. KM thanks the Space Astronomy Summer Program (SASP) hosted by STScI. The authors thank STScI staff who participated in this study.

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

# A    Appendix: System Description Details

Our RAG system follows the framework outlined in Wu et al. (2024). In early trials, we included the full-text paper as context, but this frequently resulted in incorrect citations; the RAG-based Slack bot would often focus on references in a paper's introduction or the related works section, rather than the actual results from the paper. Therefore, we fetch just the abstracts and conclusions from each arXiv paper (which is extracted and cleaned via the arXiv LaTeX source code). We also insert the metadata for each paper into the prompt string. We only include the five most semantically similar papers (determined via cosine similarity) so as not to exceed the 128,000-token context length of the model (gpt-4o-2024-05-13).

In the code listing below, we provide the full prompt template used by the RAG-based chat bot. The full prompt concatenates the context string (context_str), which contains the arXiv ID, paper year, abstract, and conclusions section. After providing the instructions and the context string, we append the query. This prompt template resulted from multiple iterations from our team's initial tests.

```
You are an astronomy research assistant that can access arXiv paper chunks as
context to answer user queries. Some of the context may contain LaTeX code,
but please answer the query in natural language like a professional research
astronomer. Match the level of specificity or generality as the query. ALWAYS
cite ALL relevant papers using EXACTLY the citation style in the context, in
parentheses: `(<https://arxiv.org/abs/1406.2364|1406.2364>)`. The current year
is 2024. Unless directed otherwise, prioritize MORE RECENT results based on the
paper YEAR. Answer in 100 words or fewer. If the query is not related to astronomy
in any way, or if none of the papers can help you answer this, then say 'I cannot
answer'.

The arXiv astro-ph papers context string are below:
--------------------
{context_str}
--------------------

Query: {query_str}
Answer:
```

# B    Appendix: Distribution of Participants Across Demographic Categories

| Gender | | Race | | Year of PhD Completion | |
|---|---|---|---|---|---|
| Man | 25 | Asian | 5 | <2000 | 5 |
| Woman | 10 | Hispanic/LatinX | 2 | 2000 – 2009 | 9 |
| | | White | 26 | 2010 – 2015 | 8 |
| | | White; Hispanic/LatinX | 2 | >2015 | 11 |

| Career Track | | Age | | |
|---|---|---|---|---|
| Astronomer | 15 | <34 | 7 | |
| Postdoc/Fellow | 3 | 35–44 | 12 | |
| Scientist | 7 | 45–54 | 10 | |
| Self-identified Researcher | 8 | 55–64 | 3 | |
| | 8 | >64 | 2 | |

Table 3: Counts of answers selected on an optional self-reported demographic survey provided to all participants. Not all participants answered every question.

# C Appendix: Descriptions and Examples for Query Types

| | Description | Example |
|---|---|---|
| Knowledge seeking: specific factual | Questions about very specific pieces of information, such as characteristics, facts, parameters of specific objects, phenomena, or processes. | What is the most massive known spiral galaxy in the universe? |
| Deep knowledge (Including opinion or speculation) | Research-focused inquiries focusing on brainstorming, summarization, or even speculation and opinions after extensive information searches. Such queries often require deep analysis and multi-level reasoning. Examples are queries requesting detailed explanations or clarifications on specific patterns or phenomena, comparisons or evaluations between two or more entities or performance within a specific context. | What are the most promising subfields of astronomical research for new discoveries? |
| Bibliometric search: topic | Questions that search for references on a specific matter, topic, option piece, or phenomenon. Rather than broader portraiture analysis, the question is seeking specific and highly relevant references. | Please list the most recent papers on multiple populations in Globular clusters. |
| Knowledge seeking: broad description or common sense | Questions that have distinct and correct answers, but answers are broad and descriptive, such as a paragraph about phenomena and processes | What physical characteristics indicate the transition between a brown dwarf and a low mass star? |
| Stress testing | Questions that are unrelated to the astronomy, intentionally misleading, or generally try to bypass guardrails | Why are the cores of all rocky planets made of Silicone? |
| Bot capabilities | Questions that seek information on the bot's capabilities, features, scope of knowledge, or limitations. Questions are not directly related to astronomy or information retrieval, but rather seek information on how to interact with the bot and what its characteristics and capabilities are. | What sort of questions can you answer? Provide your response in the form of a bulleted list. |
| Knowledge seeking: procedure | Questions that seek an answer in terms of instructions, descriptions of the steps or flow of a specific task, a broader picture, or recommendations on how a given procedure should be performed or how it can be interpreted, analyzed, and applied. | How can compressed sensing be used for image deconvolution in radio interferometry? |
| Bibliometric search: specific paper or author | Questions that request details, analysis, or otherwise refer to a specific paper or author (often by providing the name of the author, title of the paper, or link to the paper). | Can you list 3 papers published by <redacted: name>? |
| Unresolved topic | Questions about topics that are controversial, don't have a single clear answer, or are generally unsolved. | Does TRAPPIST-1b has an atmosphere? |

Table 4: Descriptions and examples for query types as determined through inductive coding. Query types are ordered for consistency with Table 1.

# D    Appendix: Variations in Question Type Frequencies

In Figure 3 and Figure 4, we show bar charts on the frequency of question types by the gender and the year of PhD completion of the participants.

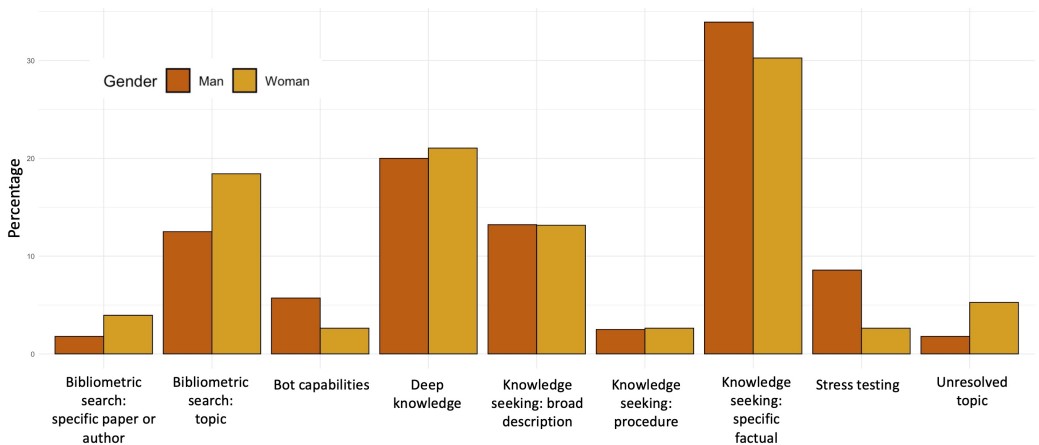

Figure 3: Frequency of question types by gender for men (24) and women (10), as self-reported on an optional demographic survey. The Y axis denotes the percent of questions asked by people of the specified gender that were coded as that type. We report percentages rather than raw counts due to data imbalance. Distributions are remarkably similar.

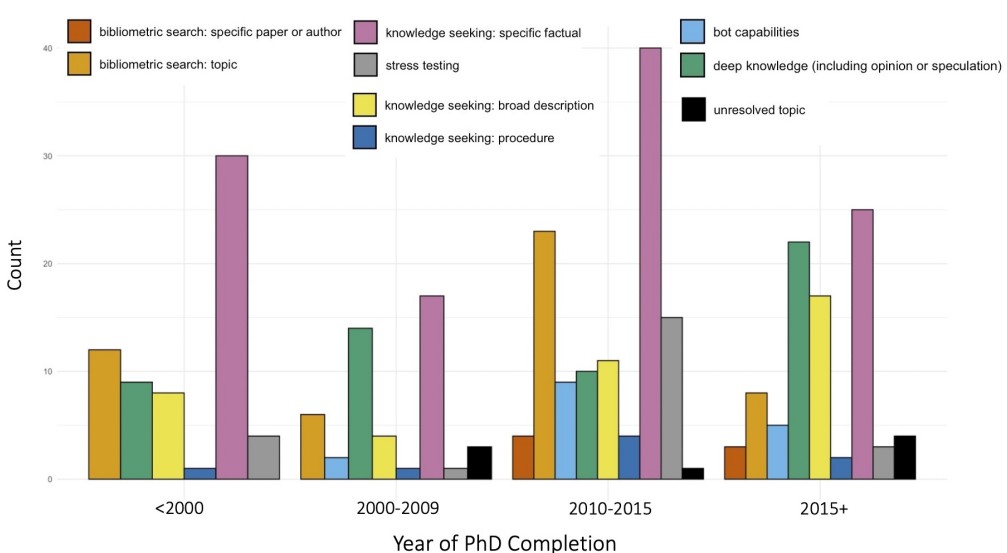

Figure 4: Frequency of question types by the year of PhD completion, as self-reported on an optional demographic survey. The number of participants in each category is reported in Appendix B. There is some variance in frequencies of question types. For example, stress testing questions were asked more frequently by people who completed their PhDs after 2010.

Figure 3 shows a bar chart of question types split by gender. The y-axis depicts the percentages of questions for each question type (assorted into our inductive codes). The gender-disaggregated question type frequencies are generally similar, with small differences only in frequencies of *Stress Testing* and *Unresolved Topic* questions.

Figure 4 shows a bar chart of question types split by year of PhD completion. The x-axis labels specifically are $< 2000, 2000 - 2009, 2010 - 2015$, and $2015+$. There are more *Stress*

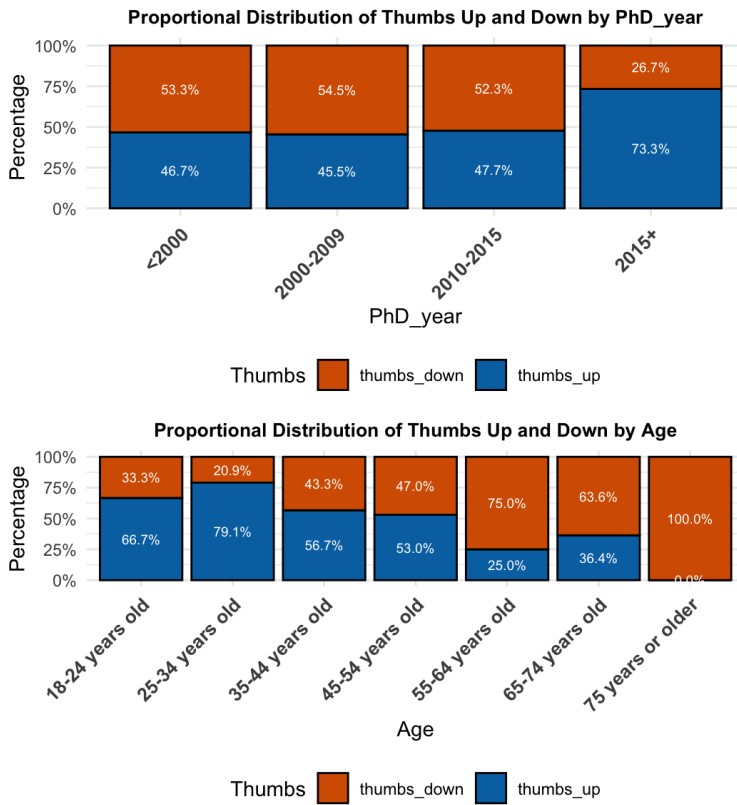

Figure 5: Distribution of how users rated the chat bot separated by year of PhD completion and age. Younger users seemed to rate the bot more positively than older users.

*Testing* and *Bot Capabilities* questions in the 2010-2015 and 2015+ groups. *Knowledge-seeking* questions are generally evenly distributed.

Figure 5 shows the distribution of how users rated `@Ask-astro-ph` by year of PhD completion and by age. Users who graduated from the PhD more recently (e.g., 2015+) and younger users (e.g., $\leq 45$ years old) rated the bot more positively.

# E   Appendix: Temporal Evolution of User Queries

Figure 6 shows the distribution of user queries by the ID for each user and the date of the query. The x-axis shows the ID for each user and the Y-axis shows the general date that queries were asked to the bot. The colors of the points are related to each different question type as seen in the legend to the right. Each individual user asked a variety of questions throughout the time period.

# F   Appendix: Sample Benchmark Application

A large issue in studying LLMs that has arisen recently is the fact that it is quite difficult to understand how to properly evaluate the quality of responses. How can researchers quantitatively score nuanced answers to complex topics with multiple correct or debated results? Using the gold benchmark dataset, we were able to compare responses from the chat bot with the expert-curated answers. We used a separate LLM to score the similarities of each chat bot response against the gold benchmark answer, and returned a relevance score between 0 (irrelevant to the gold sample) and 1 (perfectly matching the gold sample).

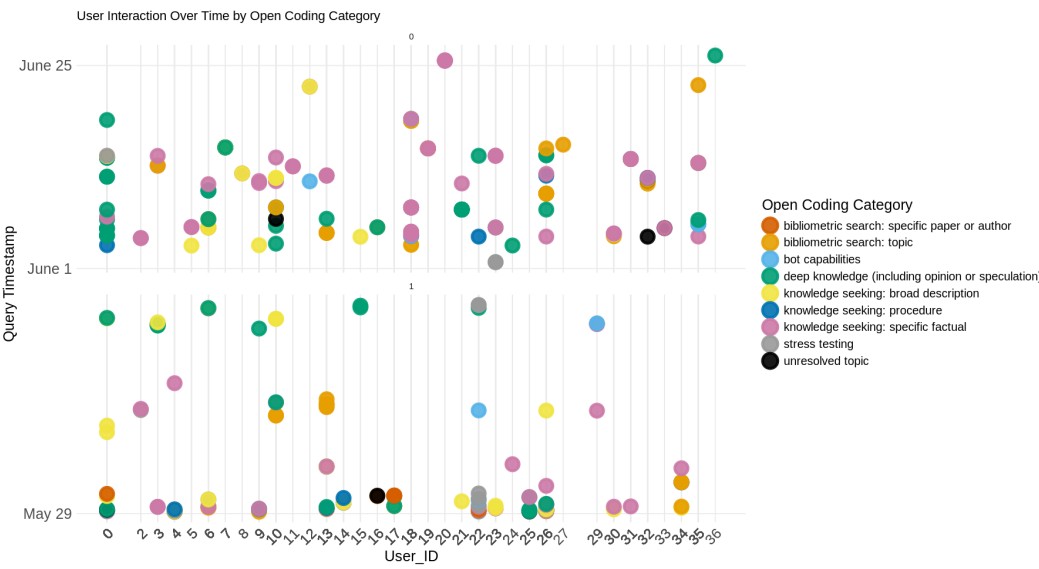

Figure 6: Frequency of question type over time for each user. Each user asks a variety of question types throughout their interactions and there is large variance by user in the distribution and ordering of question types.

An example usage of this gold benchmark dataset in this study was making judgments about the quality of two different chat bot system responses. After we initially concluded collecting user data over the course of four weeks, we internally created and tested a second upgraded chat bot with an improved retrieval system, prompts, and other adjustments. The new system is described in (Iyer et al., 2024). Using the gold benchmark dataset, we compared responses from the initial and upgraded chat bots. The relevance scores provided by the LLM judge provided us with accurate and useful numbers that were able to be plotted and analyzed. On average, the updated chat bot responses scored 0.2-0.3 points higher than the original chat bot system, allowing us to be able to quantify the quality of LLM-generated responses without needing to be an astrophysics expert.

The prompt that was used to generate these relevance scores is included below. The model used was `gpt-4o-mini`.

```
    Task: Evaluate the relevance and quality of a response compared to a gold
standard response.
        Original Query: {query_str}
        Gold Standard Response: {gold_response}
        Response to Evaluate: {response}
        Instructions:
        1. Compare the content and meaning of the two responses.
        2. Consider the following aspects:
            - Accuracy: How factually correct is the response?
            - Completeness: Does it cover all key points from the gold standard?
            - Relevance: How well does it address the original query?
```

```
        - Coherence: Is the response well-structured and logical?
        - Conciseness: Is the information presented efficiently?
    3. Assign a relevance score from 0 to 1, where:
        - 1.0: Perfect match in content and quality
        - 0.8-0.9: Excellent, with minor differences
        - 0.6-0.7: Good, captures main points but misses some details
        - 0.4-0.5: Fair, partially relevant but significant gaps
        - 0.2-0.3: Poor, major inaccuracies or omissions
        - 0.0-0.1: Completely irrelevant or incorrect
    Provide your relevance score as a float between 0 and 1.
```

