# OpenReview forum: "From Queries to Criteria: Understanding How Astronomers Evaluate LLMs"
_colmweb.org/COLM/2025/Conference — COLM 2025_

### Official Review · Reviewer_Sraf · 2025-05-02

**Rating:** 8
**Confidence:** 3
**Ethics Flag:** 1

**Summary:**

This paper conducts an in-depth study into the interactions between astronomers and LLM bots. The bot is hosted on a slack channel where the users (astronomy PhDs) can query the bot and evaluate the response via emoji reactions and comments. The bot is a simple RAG pipeline that retrieves paper abstracts from arXiv and use GPT-4o to generate the answer and related citations.
The main focus of the paper is to gain a better understanding of what types of questions do astronomers ask and how do they judge responses. Thus, the authors conducted in-depth analysis into the types of user queries, categorizing them into concrete types (e.g., knowledge seeking, stress testing, etc.).
Then, the authors conducts user studies that reveals how experts use and evaluate the responses. These qualitative results lead to recommendations to the community on how to build better evaluation settings.
Finally, the collected queries are used to construct a new benchmark.

**Questions To Authors:**

- What is the Year of PhD completion breakdown between 2016-2020 and >2020?
- What are the demographics and qualifications of the 7 astronomers who wrote the answers in the benchmark?
- Are the expert-written answers in the sample benchmark cross-validated?

**Reasons To Accept:**

- The paper is well motivated—it studies the important setting of how domain experts may interact and use the latest LLM systems. The paper chooses a suitable setting for studying this.
- The qualitative analyses are extensive and insightful given the difficulty of getting a large number of domain experts. Despite being limited to the field of astronomy, it is likely that many of the takeaways and insights can be applied to many other fields. In particular, the categorization of the queries and different aspects of how experts evaluate the response can inform future work.
- The analyses and details over the breakdown across different demographics are insightful—the difference between PhDs with varying number of years of experience motivates for diverse evaluation.
- The resulting artifact—the sample benchmark—can be useful for evaluating LLMs, though further validation and additional evaluation framework may be necessary.

**Reasons To Reject:**

- Although the proposed sample benchmark can be useful to the community, it could benefit from further validation of the evaluation framework. For example, it may be more insightful to break the evaluation into multiple aspects as listed in the Table 2 (e.g., correctness of citations vs. matching between responses and citation) as opposed to putting everything into just one number.
- The main metric for evaluating the RAG system is via the thumb up and thumb down reactions to the responses, but it is not clear how reliable these reactions are. For example, what are the agreements between different users on the same response? From the later analysis, it’s clear that the quality of the responses cannot be condensed into one of two reactions. Further expert validation could be insightful.
- The evaluation is limited to one specific system. While it is understandably difficult to evaluate multiple systems due to the human effort, it would be incredibly useful to have some comparisons between different systems; for example, one could provide two responses and ask the user to judge which one is between.

Minor comments:
- Missing related work for discussions:
    - OpenScholar: Synthesizing Scientific Literature with Retrieval-augmented LMs (Asai et al., 2024)
- Appendix A: “128,000-character context length of the model” should be “128,000-tokens”

---

> ### Author Response · Authors · 2025-05-31
>
> Thank you so much for your helpful feedback and appreciation of our work. Regarding reasons to reject (RtR):
>
> - RtR 1: We completely agree that evaluation criteria should involve multiple dimensions, as we specify in Table 2. As our work focuses on understanding human evaluation practice, we do not fully implement automated metrics for all of our proposed criteria, though we offer suggestions of how to do so in Section 5, which we encourage future work to implement
> - RtR 2: User judgments were highly correlated: out of 222 queries that received a reaction, there were only 5 cases where one user gave a thumbs up and another user gave a thumbs down. As our primary goal is not evaluating this specific deployed system, but rather understanding user evaluation practice, we do not provide finer-grained evaluations of this system. Instead we propose user-grounded guidelines for future evaluations.
> - RtR3: Thank you for this suggestion. We agree that comparing outputs of different systems is generally useful. Our work suggests that users should rate outputs along a range of criteria rather than just selecting which one is preferred.
>
> Thank you for the minor comments, we will address them in the revisions.
>
> Questions:
> - Q1: 7 participants received PhDs in 2015-2020 and 4 received PhDs after 2020.
> - Q2: 2 of the annotators are senior astronomers (10+ years post-PhD), 3 of them are early career astronomers (postdocs or assistant professors), and 2 of them are senior astronomy PhD students. They currently hold primary appointments at 3 different institutions, and all of them have secondary appointments and/or completed training at additional institutions. They focus on a variety of subfields within astronomy (e.g., we recruited an additional annotator when we realized none of our original 6 focused on exoplanets). One of the annotators is a women and 6 are men.
> - Q3: One of the astronomer authors on our paper read through all of the gold responses and validated that they were reasonable. They did observe that the responses varied in style, which is likely reflective of varying user preferences

---

> > ### Comment · Reviewer_Sraf · 2025-05-31
> >
> > Thanks for the response. Although I still believe the study could benefit from more systems, I don't have any other major concerns or questions for now. I will maintain my positive score.

---

### Official Review · Reviewer_6qMG · 2025-05-11

**Rating:** 3
**Confidence:** 5
**Ethics Flag:** 1

**Summary:**

**Paper summary:** The paper presents the results of an empirical study where astronomers were given access to retrieval-based LLM summaries of user queries. Specifically, users were able to query a bot through Slack, which would retrieve relevant papers using similarity with pre-computed embeddings of a fixed set of papers. Relevant papers were then included in the context of a proprietary LLM to generate responses to user queries. The paper studies (1) what kinds of queries the users used, and (2) how they evaluated the responses, based on post-study interviews.

**Pros:** The paper is clear and well-written, using a good mix of tables and figures for presentation. I also appreciate the focus on a domain science -- astronomy -- at a time when domain scientists are trying to understand how to best leverage LLMs for research. I think this is the main dimension along which the paper spikes.

**Cons:** Methodologically, the study described in paper is somewhat weak, to the extent that it's unclear what conclusions to actually take away. As the authors do acknowledge, the sample size of the study is quite limited, with the final response evaluation criteria based on 11 interviews at the specific institution the study was conducted in. It is unclear how the participants themselves approached the bot during the study -- e.g., as an object to stress-test (as indicated by some queries), or as a tool for serious research, confounded by the fact that they were being "watched". The tools used in the study themselves are somewhat outdated -- using RAG embeddings of a small set of papers up to July 2023. In an era where LLMs can conduct large-scale searches not limited to a fixed set of papers, it is unclear whether the study actually reflects real-world use cases. Finally, the authors introduce a "golden" benchmark for evaluating LLM responses to astronomy queries. However, no details on the benchmark are included in the main content of the paper, or how in practice it can be used as an evaluation tool for LLMs, so I am not considering it in this review.

**Questions To Authors:**

- Given concerns about generalizability, the tech stack (fixed set of papers, no up-to-date knowledge), and user intent, to what extent would you expect the study to accurately reflect modern and forward-looking uses cases of LLMs in astronomy?
- How did you control for the so-called Hawthorne effect (i.e., participants modifying behavior because they were being observed)? Can you distinguish between users who were genuinely using the bot for research versus those who were stress-testing it because they were participating in a study?
- Benchmark details: how can the "golden" benchmark mentioned in the paper be used practically, e.g. to evaluate current general-purpose LLMs?
- Are there variations across career stage that may be relevant in the study? (I would think this is particularly interesting to understand adoption patterns)

**Reasons To Accept:**

- Clear and well-written paper, with good presentation;
- Application to a quantitative domain science -- astronomy -- which is well-suited for many LLM use cases.

**Reasons To Reject:**

- The study is based on only 11 interviews from a single institution, severely limiting generalizability of findings to the broader astronomy community or other scientific domains. With only 35 users generating 367 queries over 4 weeks, the dataset is too small to draw robust conclusions about query patterns or evaluation criteria for astronomical research;
- The study doesn't adequately control for how participants approached the task -- whether they were genuinely using the bot for research or merely stress-testing it due to being observed, which confounds the interpretation of results;
- Outdated stack: The system uses pre-computed embeddings from papers up to July 2023 with fixed paper sets, which doesn't reflect current LLM capabilities for dynamic, large-scale literature search. Given this, it's unclear whether it reflects practical real-world use cases;
- While the paper introduces a "golden" benchmark for evaluation, it provides no implementation details or demonstration of practical utility in the main paper.

---

> ### Author Response · Authors · 2025-05-31
>
> Thank you for your review of our paper. We address your Reasons to Reject (RtR) and Questions to the Authors in a combined manner as follows:
>
> - RtR 1: Our study is based on a deep analysis of 367 queries from 35 users with 11 follow-up interviews all of whom were recruited with expertise in astronomy. If we expanded the scope, we would need to reduce the depth of analysis, which currently includes manual coding of user queries, follow-up interviews, and construction of gold query-answer pairs. Furthermore, our study matches or exceeds the scope of similar human-centric studies, where a sample size of 6-20 users is common (e.g. [Mirowski et al. 2023](https://tinyurl.com/58j55jp5)  similarly studying LLM usage with 15 users total; [Lazar et al. 2017](https://tinyurl.com/ye25ebjv) discuss methodology more generally; other example studies include [Feng et al. 2025](https://tinyurl.com/ymbdt9ur), [Zhao et al. 2022](https://tinyurl.com/5fcrkdh5) and [Liao et al. 2020](https://tinyurl.com/4unmx5au)). As noted by other reviewers, the scope of our study is sufficient for identifying interesting trends in user evaluation practice that can help improve LLM evaluation and can be further validated in follow-up replication studies.
> - RtR 1, RtR 3 and Q1 about “outdated tech stack”: Our deployment and data collection was conducted in June 2024. Thus the underlying set of papers ending in July 2023 excludes only the most recent year of research. Astronomy research moves more slowly than ML, and the exclusion of the most recent year of data is extremely minor compared to the large volume of literature. In our review of user queries, we identified only a very small number (~5) of queries that targeted topics too recent for our system. Furthermore, our work focuses on analyzing human evaluation practice: what types of queries users posed and how they judged responses, not evaluating or optimizing the performance of the specific system we deployed.  Our system was sufficient for this use case. There is also clear evidence that users found the system useful. In follow-up interviews, they appreciated that our setup enabled providing specific references (Sec 4.2) and they continued querying it throughout the duration of the study (Fig 6; Appendix F).
> - RtR 2/Q2: We would like to clarify a misconception: our study does not assume that users were genuinely using the system for research. We assume that users were primarily seeking to test and evaluate the bot. This assumption is validated through follow-up interviews, where almost all interviewees described probing the bot about things they knew (Sec 4). A smaller subset of users did try out the bot on actual use cases, but primarily ones they were also approaching through other means (Sec 4). The assumption that most queries were about evaluation rather than usage is integrated throughout our work, including its title. The Hawthorne effect is primarily a concern for researchers studying behaviors that are not socially desirable, which is not the case with our work, especially since prior studies of LLM usage suggest users do not modulate their queries (e.g. they disclose private information Mireshghallah et al. 2024). Attempting to mitigate any potential Hawthorne effect by not informing users that we are logging their queries would be unethical.
> - RtR 4/Q3: Sec 5.2 details the construction of the benchmark and validates its usefulness for evaluating LLMs. We show that the automatically estimated closeness between model generated responses and gold responses are highly correlated with human judgements (Pearson correlation = 0.8239). Could you please clarify what details about the benchmark you expected that are not already included in 5.2, or update your review if you originally missed this section?
>     We also have used the gold benchmark data to compare our original chatbot system with an updated system constructed after the study’s completion. We did not include these results in the original submission as we considered them beyond the scope of this work, but we will add them to the appendix.
> - Q4: We greatly appreciate your interest in the results and ways this work can contribute to a broader understanding of LLM usage in its current scope. We report the frequency of queries by number of years since PhD completion (a reasonable proxy for career stage) in Fig 4 in Appendix D. We do not see consistent patterns in how total volume of usage varied. There is some variance in query type: stress testing queries were asked more frequently by people who completed their PhDs after 2010. There is a striking pattern in ratings of bot responses: there are more positive ratings on responses to queries asked by younger participants who completed PhDs later (Fig 5, Appendix D, discussed in main Sec 6). This result may indicate unequal adoption and greater familiarity with AI among earlier career researchers, though we do not further validate this result as it is not the focus of our work.

---

> > ### Comment · Reviewer_6qMG · 2025-06-02
> >
> > I thanks the authors for their response, and considering my review. I in particular appreciate the point, "We would like to clarify a misconception: our study does not assume that users were genuinely using the system for research.", which indeed addresses many of the points originally brought up. As I understand now, the paper does not attempt to understand how domain scientists use LLMs for research, but rather how they evaluate LLMs when asked to do so.
> >
> > With the premise clarified, my concern shifts somewhat, as I'm not entirely certain how studying evaluation behavior from domain scientists under controlled conditions itself is a question from which one can take away practical learnings. E.g., I can see how understand how astronomers use LLMs for research can help develop better benchmarks, and improve LLM capabilities and scaffolding -- however, the same isn't clear if understanding evaluation behavior is the goal.
> >
> > I also thank the authors for clarifying about the benchmark (I assume you mean Sec. 5.1, rather than 5.2?). Is it correct that, in order to test an external model on this benchmark, one would take its response, together with the golden response, and ask a third party model (4o in this case) to compare them on various axes? Again, I'd be curious what practical takeaways could be made from such a benchmark -- e.g., if 4o vs a Gemini model score differently on the constructed benchmark with LLM judge, does that say something about the research usefulness of these models, or their use as evaluation artifacts (as the paper premise)?

---

> > > ### Author Response · Authors · 2025-06-04
> > >
> > > Thanks so much for your engagement on our paper.
> > >
> > > Regarding why we study evaluation practice:
> > >
> > > To build useful LLMs for scientific domains, we must understand *what domain experts actually value in model outputs*. Our study distills how astronomers (highly trained experts) judge LLM responses: what criteria they care about (4.2) and how they probe for it (4.1). Their queries also reveal specific examples of questions domain experts expect usable models to be able to answer. Existing benchmarks often rely on crowdworkers, synthetic questions, or questions from standardized tests—approaches that don’t necessarily reflect expert judgment. The insights we derive from analyzing user behavior are actionable: they directly inform better evaluation metrics and can also be applied to data collection strategies, and even model training objectives.
> > >
> > > We’d also like to clarify that our experimental design was a close-to-realistic setting rather than a tightly controlled setting (e.g. we did not assign users tasks in a fixed time frame like Wang et al. (2024)) and we did not specifically direct them to evaluate the bot. Instead we invited them to interact with it on their own time, encouraged them to try using it in their research, and requested they provide feedback through ratings and comments. In practice, we observed that most people’s use was evaluative (confirmed through follow-up interviews), with some queries inspired by actual use cases, and so we constructed our analysis accordingly.
> > >
> > > Regarding the benchmark:
> > >
> > > Yes, we meant 5.1, our apologies. Your description of how we demonstrate one use case is correct. The high correlation between human ratings and GPT-4o-mini judgments suggests that this type of benchmark is able to capture when models output better-quality responses. Our experiment directly shows that the benchmark can aid in identifying which questions (e.g. topics or types of queries) a model performs poorly on, requiring further development. While we do not test this specifically, we expect the benchmark will also be useful for comparing models: e.g. if model A outputs responses more similar to the gold benchmark responses than model B, we expect users to prefer model A.

---

> > > > ### Comment · Reviewer_6qMG · 2025-06-07
> > > >
> > > > Thank you for your thoughtful response! I appreciate the clarification and confirmation about how the benchmark could be used, but am still concerned about a fundamental framing disconnect.
> > > >
> > > > You state that "our study does not assume that users were genuinely using the system for research," yet you also mention that you "invited them to interact with it on their own time, encouraged them to try using it in their research" and describe the setting as "close-to-realistic.", which seems somewhat contradictory. If the primary behavior was evaluation and stress-testing rather than genuine research use, can we be confident that the criteria identified (e.g., citation quality, hedging, specificity) reflect what domain scientists value when actually using LLMs for research versus what they check when testing a system in an evaluative setting?
> > > >
> > > > When positing that "to build useful LLMs for scientific domains, we must understand what domain experts actually value in model outputs," it's unclear why evaluation behaviors in an artificial testing context would reveal authentic research values. The claim that insights from analyzing primarily evaluative interactions will "directly inform better evaluation metrics" for research use seems to require an assumption that evaluation criteria remain constant across contexts, which is not validated. For example, it is not clear how to use the fact that a subset of users tried to "trick" the LLM when evaluating it, to build better models.

---

> > > > > ### Author Response · Authors · 2025-06-07
> > > > >
> > > > > Thanks for your perspective. In interviews with our participants, we do discuss how users might expect to use LLMs for research. Before asking participants about queries they sent during our study and criteria they care about, we asked, “What do you want out of a bot like this?” and “How might you expect to use it?” When we asked about their queries, participants did describe ways in which their queries helped assess properties they would want in a usable system (evidence of this is in user quotes in 4.2, with many others omitted for brevity). We do see your point that users’ may not have anticipated all of the criteria that they would care about, especially since only a subset described trying the bot on real use cases (4.1.2). We’ll add as a limitation in the main text of the paper that it is possible for users’ desired criteria to shift over longer-term use of a system integrated in their workflow.
> > > > >
> > > > > We greatly appreciate your engagement in discussion. We’d like to highlight the broader perspective on our work that may be getting lost in the back-and-forth. Numerous current LLM benchmarks for science are far-removed from user perspectives and realistic use cases (e.g. many draw questions from standardized tests). While we can’t guarantee that properties desired in LLM systems will never change, our grounding of evaluation criteria in actual user interactions and interviews is an enormous improvement over work that offers no justification for evaluation criteria. We also can’t think of a better way to elicit what users might want in an end system than inviting them to interact with an initial version and asking them about their experience (which is why this is standard HCI methodology).  As noted by reviewers g8n8, aurw, and Sraf, our work offers valuable insight that would be useful to LLM researchers and developers if published.

---

### Official Review · Reviewer_aurw · 2025-05-12

**Rating:** 7
**Confidence:** 5
**Ethics Flag:** 1

**Summary:**

This paper presents an interesting study on how researchers (in this particular case, astronomers) use LLMs for their research activities. The authors deployed a GPT-4o-based slackbot and recorded the interactions between researchers and the bot. Behind the bot is a RAG system that is connected to a database of astronomy papers. The authors focused on answering two questions: in a realistic research pipeline, (1) what types of questions people ask and (2) what criteria people use to judge the bot. Answering these questions involves careful manual annotations of the questions and extensive interviews with the participants. Based on the findings, the authors made some recommendations on how to build a benchmark with realistic and indicative criteria, and collected some expert-annotated answers to form a high-quality astronomy datasets for researcher-LLM interactions.

**Questions To Authors:**

Please see reasons to reject. One question: why use gpt-4o-mini for the human/LLM judgement correlation experiment? Why not use a more powerful model?

**Reasons To Accept:**

This study is interesting and valuable in multiple aspects:

(1) The data it provides (the collected questions, question type annotations, participants' reactions to LLMs' answers, interviews with participants, expert-annotated answers) is extremely valuable for LLM developers, evaluation researchers, and domain experts who intent to use LLMs.

(2) The data quality is very high: the authors described in details how they resolve annotation conflicts and ensure expert-written answers are high-quality. The description of the whole experiment pipeline (slackbot deployment) is also very clear.

(3) The discussion around "good criteria" is interesting: instead of just arguing certain perspectives are important (which a lot of prior work has done), the authors interviewed participants and summarized their opinions (as real users who tried out the system). It is certainly very inspiring for future evaluation frameworks.

**Reasons To Reject:**

(1) Lack of discussion on existing (and future) evaluation mechanisms. There have been several works on this type of citation-based question answering LLM evaluation, such as [Gao et al., 2023](https://arxiv.org/abs/2305.14627) and [Asai et al., 2024](https://arxiv.org/abs/2411.14199). They discussed a similar application scenario and also detailed automatic evaluation metrics that could reflect user perspectives. The discussion on criteria in this paper is interesting, but not concrete.

(2) Limited scope: the paper mainly focuses on Astronomy. The interview is limited to 11 participants and the final dataset only has 42 question-answer pair.

That being said, I still believe the paper presents some very interesting observations and data to the LLM community, and the above shouldn't become reasons to reject.

---

> ### Author Response · Authors · 2025-05-31
>
> Thank you very much for your helpful feedback and your appreciation of the value of our work.
>
> Regarding reasons to reject (RtR):
>
> - RtR 1: We discuss comparisons with existing evaluation benchmarks in Section 2 and 5, and in revisions we will add more detailed comparisons with [Gao et al. 2023](https://arxiv.org/abs/2305.14627) and [Asai et al. 2024](https://arxiv.org/abs/2411.14199) (Asai et al. 2024 was released after our data collection and analysis had been completed). These two papers propose methods to automate evaluation criteria, whereas our work focuses on discovering what criteria users care about and how they enact them. Thus our work is complimentary in that it validates the usefulness of metrics like correctness and coverage, uncovers additional criteria like specificity and clarity, and further shows that certain types of queries need to be included in benchmarks to surface these properties.
>
> - RtR 2: Our full analysis data set consists of 367 queries from 35 users with diverse backgrounds (appendix B). While we agree that expanded scope would be valuable, the level of depth that we conduct would not be feasible with a substantially larger scope. We would gladly provide support for follow-up replication studies that repeat our work in a different domain (to facilitate this, the code for our slackbot is already public, though not linked in the paper for anonymity), and we also agree that our work in its current form offers substantial contributions that are of interest to LLM researchers.
>
> Question to authors:
>
> - We initially conducted the human/LLM judgement correlation experiment with `gpt-4o-mini` because as an inexpensive model, it is more practically usable by researchers conducting multiple rounds of evaluations. As we already saw high correlation with human judgments (0.8239) with `gpt-4o-mini`, we did not conduct further experiments with more powerful models.

---

> > ### Comment · Reviewer_aurw · 2025-06-02
> > **Thank you**
> >
> > Thank you for your response! The work is really interesting and all the analysis is very valuable. I'll keep my original score.

---

### Official Review · Reviewer_g8n8 · 2025-05-12

**Rating:** 6
**Confidence:** 3
**Ethics Flag:** 1

**Summary:**

The authors present an exploratory study into how expert users of a domain-specific search engine query, interact and assess the RAG-based system that has been deployed. The two underlying research questions aim to answer (a) what types of questions users (in this case astronomers) submit to the system and (b) how users actually judge the quality of the system. Topically this is a great fit as the work addresses a range of key topics spelled out in the Call for Papers.

**Questions To Authors:**

* More of a suggestion than a question: I get the impression that the work resembles a specific use case of “professional search” which is distinctly different from other types of search (such as web search). I would contextualise this as part of the related work (a good recent reference could be the chapter on Professional Search by Suzan Verberne in the recently published IR book by Alonso and Baeza-Yates: https://dl.acm.org/doi/10.1145/3674127.3674141)
* Is there some evidence that the arXiv collection is representative for what astronomers use as their first point of call? After all these are papers that have not been peer-reviewed.

**Reasons To Accept:**

* The paper is not just well-written but the authors also provide a detailed appendix that aims at making the work as reproducible as possible which is a major strength.
* Reproducibility and replicability have also been taken into account in the Methodology section. Despite the shortcomings (see below) the authors have done a great job here.
* The discussion of limitations is compelling (but the page limit has been violated by placing it on an extra page beyond the imposed 9-page limit).
* The created benchmark will be an interesting asset for the community (although it is very small overall).

**Reasons To Reject:**

* The work feels more like a pilot study that should be followed up by a more systematic investigation. All the investigations and findings are very interesting but it does not become clear how generalisable they are which is due to a number of reasons including (a) a very small number of queries, (b) concerns around the representativeness of the setup (including dataset, users, questions etc.) and (c) the lack of a plausible baseline which would allow a comparison between the proposed architecture and how users currently conduct their research. Among other things such baseline would provide a very important alignment between what the RAG-based system might offer vs. what a standard (strong) baseline offers already.
* From a community perspective, how representative are the recruited users for the community as a whole (or for a specific downstream use case)?
* There is a lack of contextualisation with related studies in Section 3.2 but also in parts of Section 4. What other studies have been conducted (in other communities and contexts) that can be compared to this work. Clearly inductive coding is a method that has been used widely. Pros and cons of different alternatives should be discussed, and the results should be compared and contrasted with previous studies. Just to pick an example: Krippendorff’s alpha is very low making generalisable insights difficult. How does that compare to other studies? Section 4.2 is better in this respect in that some related work is being referenced.

---

> ### Author Response · Authors · 2025-05-31
>
> Thank you for your helpful feedback and appreciation of the strengths of our paper, including its fit for the conference, that its “investigations and findings are very interesting”, and has high quality writing, which facilitates reproducibility and follow-up work.
>
> Regarding reasons to reject (RtR):
>
> - RtR 1: Our study matches or exceeds the scope of similar human-centric studies that focus on deep analysis of user behavior. Our study contains 35 users with 11 follow-up interviews, while a sample size of 6-20 users is common (e.g., [Mirowski et al. 2023](https://dl.acm.org/doi/pdf/10.1145/3544548.3581225) similarly studying LLM usage with 15 users total; [Lazar et al. 2017](https://www.sciencedirect.com/science/article/abs/pii/B978012805390400008X) discuss methodology more generally; other example studies include [Feng et al. 2025](https://dl.acm.org/doi/10.1145/3706598.3713139), [Zhao et al. 2022](https://aclanthology.org/2022.naacl-main.24/) and [Liao et al. 2020](https://dl.acm.org/doi/abs/10.1145/3313831.3376590)). While our analysis focuses on one domain and a limited set of users, it is extremely deep and multi-faceted. Expanding scope would force us to reduce the depth of analysis (we already had to drastically cut insights from interviews to fit the page limit; we would not be able to manually analyze all queries). We agree that replicating our study with a different user group or investigating how specific aspects scale to larger data are great directions for follow-up work. In its current form, our work already offers valuable insights (e.g. as noted by Reviewer aurw and Sraf) that would be useful to other researchers if published.
>
>     We do not compare against a baseline because our project focuses on exploratory analysis of user behavior, rather than testing performance of a specific system. It is unclear to use how comparison against a baseline could improve our understanding of evaluation practice.
>
> - RtR 2: Our study participants are diverse in terms of race, age, career track, and phase of career (Appendix B, Table 3), and they have varying specialities within astronomy (exoplanets, galaxy clusters etc.) as demonstrated through their queries. Thus, our users reflect a broad range of types of astronomers. Additionally, the dispersion of query types across users (Fig. 2) and overlap in content in follow-up interviews (Section 4) with diminishing findings from new interviewees suggest that our user pool was sufficient to achieve saturation.
>
>     There is one main limitation of our study population: that researchers entirely disinterested in AI (with neither strong positive nor negative opinions) likely did not choose to participate. However, this is a fundamental limitation of research studies conducted ethically with informed consent. Other limitations include our focus on professional astronomers who already have PhDs, rather than students, and people currently in the U.S. (though many participants are not originally from the U.S.), but we do expect our population to be representative within these criteria. We will include these limitations in the main text of the paper.
>
> - RtR 3: We primarily discuss comparisons with other studies in Related Work (Section 2). We will expand this section, including incorporating the references in RtR 1 and integrating them in 3.2 and 4. In general we follow standard best practices for inductive data coding and interview studies. While some prior work has considered aspects of LLMs to evaluate (4.2), less work has examined what types of queries users pose in order to probe for this criteria (4.1). Because our work is novel in this aspect, there are fewer relevant works for this section.
>
> Questions to authors:
>
> - Q1: Thank you for the references to professional search. We will incorporate them.
>
> - Q2: arXiv is heavily used by astronomers. Surveys have shown 93-94% of physicists and astronomers use arXiv with 100% usage in astrophysics ([Narayana and M.K. Bhandi 2022](https://baas.aas.org/pub/2022n2i016/release/1)). For this reason, arXiv has also been used as a data source in constructing AI for astronomy in prior work ([Perkowski et al., 2024](https://iopscience.iop.org/article/10.3847/2515-5172/ad1abe)).

---

> > ### Author Response · Authors · 2025-06-06
> > **Looking forward to your response**
> >
> > Dear Reviewer g8n8,
> >
> > Thank you again for your time and thoughtful feedback on our paper. As the discussion period is ending in a few days, we'd like to kindly remind you to review our latest response and please let us know if you have any remaining questions or concerns.

---

> > ### Comment · Reviewer_g8n8 · 2025-06-07
> >
> > I appreciate the effort by the authors to address my concerns. The additional references that have been incorporated to back up the experimental setup and general design decisions are more compelling than the original text. I will up my score and now consider this contribution to be stronger than before. A better contextualisation of related work is still missing but is promised to be included by the authors. It is also assumed that the argumentation included in the rebuttal will be incorporated in a revised version of the paper.

---

### Comment · Program_Chairs · 2025-04-03

This paper violates the page limit due to adding a limitation sections beyond the page limit. COLM does not have a special provision to allow for an additional page for the limitations section. However, due to this misunderstanding being widespread, the PCs decided to show leniency this year only. Reviewers and ACs are asked to ignore any limitation section content that is beyond the 9 page limit. Authors cannot refer reviewers to this content during the discussion period, and they are not to expect this content to be read.

---

### Decision · Program_Chairs · 2025-07-08

**Decision:**

Accept

**Comment:**

This paper investigates how astronomers interact with and access a RAG system built on a domain-specific retrieval component. The topic is a strong fit for COLM, and I believe both the study and the collected dataset will be highly valuable to the community. I consider the paper worth publishing at COLM and recommend its acceptance.

I do not see any major concerns that would justify rejecting the paper:
* The study is relatively small in scale (11 interviews, 35 users generating 367 questions), but I acknowledge the challenge of expanding the scope given the need for deep domain expertise.
* The paper could benefit from a discussion on how the evaluation criteria might be automated, along with references to relevant work, though this can reasonably be left for future work.

I encourage the authors to address these points carefully in the final version.